# The Definition of the Best Margin Cutoff and Related Oncological Outcomes After Liver Resection for Hepatocellular Carcinoma: A Systematic Review

**DOI:** 10.3390/cancers17111759

**Published:** 2025-05-23

**Authors:** Abdallah Al Farai, Federico Sangiuolo, Dana Albaali, Mahmoud Ajoub, Fabio Giannone, Gianluca Cassese, Fabrizio Panaro

**Affiliations:** 1Surgical Oncology, GI Program, Sultan Qaboos Comprehensive Cancer Care & Research Center, University Medical City, Muscat 123, Oman; a.alfarai@cccrc.gov.om (A.A.F.); d.albaali@cccrc.gov.om (D.A.); m.ajoub@cccrc.gov.om (M.A.); 2Division of Hepato-Pancreato-Biliary, Oncologic and Robotic Surgery, Azienda Ospedaliero-Universitaria SS, Antonio e Biagio e Cesare Arrigo, 15121 Alessandria, Italy; federico.sangiuolo@ospedale.al.it (F.S.); gianluca.cassese91@gmail.com (G.C.); fabrizio.panaro@ospedale.al.it (F.P.); 3Department of Research and Innovation (DAIRI), Azienda Ospedaliero-Universitaria SS, Antonio e Biagio e Cesare Arrigo, 15121 Alessandria, Italy; 4Department of Health Sciences, School of Medicine, University of Eastern Piedmont “Amedeo Avogadro”, 28100 Alessandria, Italy

**Keywords:** hepatocellular, carcinoma, margin, outcomes

## Abstract

Hepatocellular carcinoma, a common type of liver cancer, is often treated through the surgical removal of the tumor. However, doctors still debate how much healthy liver tissue should be removed around the tumor to ensure the best chance of survival and reduce the risk of the cancer returning. This study reviews past research to better understand how the width of the surgical margin affects long-term patient outcomes. By analyzing different margin sizes across a wide range of cases, the authors aim to clarify whether a wider or narrower margin offers better results and how factors like tumor type or liver condition might influence the results. These findings could help surgeons make more personalized decisions for each patient, leading to more precise and effective treatment strategies in liver cancer surgery.

## 1. Introduction

Hepatocellular carcinoma (HCC) is the most common primary liver cancer in adults [1] and ranks as the fourth leading cause of cancer-related mortality worldwide [2]. In recent years, its prevalence has significantly declined, primarily due to advancements in hepatitis treatment and HBV vaccination programs [3]. However, in Western countries, HCC incidence has risen, largely driven by metabolic risk factors. The disease burden is expected to increase further due to the growing prevalence of alcohol-related liver disease and metabolic dysfunction-associated liver steatosis [4].

HCC treatment requires a multidisciplinary approach, considering both oncologic outcomes and underlying liver conditions, which are present in approximately 90% of cases [5,6]. Surgery remains the most effective treatment [7], encompassing both liver transplantation and liver resection. Due to organ shortages, liver transplantation is reserved for select patients based on their underlying liver disease. Consequently, liver resection continues to be the primary treatment for most cases [7]. To optimize outcomes, the accurate staging of HCC is crucial, taking into account tumor-related characteristics and residual liver function. The Barcelona Clinic Liver Cancer (BCLC) staging system is the most widely used for prognostic assessment and treatment planning [8].

Surgical resection is the gold standard for patients with very early- and early-stage HCC (BCLC 0-A). However, it is associated with a high recurrence rate, reaching 50–60% within three years and 70–90% within five years post-surgery, significantly affecting patient survival. Most recurrences occur within the first two years—termed “early recurrence”—which is strongly linked to poorer survival rates [9]. Simon et al. in 2018 also proposed a “very early recurrence”, when it occurs within 6 months after surgery [10]. In his study, this recurrence pattern was associated with a worse prognosis when compared with both early and late recurrence (a median OS of 20.4 vs. 41.6 vs. 36.0, respectively; *p* < 0.01), and an incomplete resection (R1) was among its main risk factors. A recurrence that occurs after 2 years of treatment is defined as “late”, and it seems related to a de novo HCC development, independent from the primary neoplasm.

Several factors have been shown to be associated with recurrence, depending on the timing and patterns of the recurrence (10.21037/hbsn-22-579.). In particular, resection margins may play a pivotal role in the very early and early recurrence risk. Previous studies have reported and proposed different optimal cutoffs to ensure the best oncological outcomes while maximizing the preservation of enough liver parenchyma. In this context, some authors have proposed the superiority of anatomical resection (AR) for HCC treatment in the light of lower local recurrence rates, that is, the resection of the entire liver parenchyma vascularized from the portal branch suppling the tumor [11,12]. However, many other studies and meta-analyses did not confirm such results, and currently there is still no consensus about the best resection margins for HCC, as well as the need for AR [13,14].

The aim of this study is to systematically review the oncological results of different margin widths after liver resection for HCC, in terms of both OS and DFS.

## 2. Materials and Methods

### 2.1. Literature Search

This was a systematic review of the literature performed according to the Preferred Reporting Items for Systematic Reviews and Meta-Analysis (PRISMA) guidelines [15]. A systematic literature search was conducted using the PubMed, MedLine, and EMBASE databases in April 2024, looking for all the articles providing the long-term outcomes of resected HCC according to margin status. Two reviewers (D.A. and A.A.F.) performed the initial literature screening to detect any potentially relevant articles, using the following combinations of terms: “Hepatocellular [Title/Abstract] AND/OR carcinoma [Title/Abstract]”, “Margin [Title/Abstract]”, AND “Outcomes [Title/Abstract]”. After this primary search, the same authors screened the articles based on the eligibility criteria. In the case of inconsistencies, a third author (F.G.) was asked to independently make the final decision. Only the studies reporting the survival and/or recurrence data of resected HCC treated with curative intent and their correlation with margin status with a well-defined cutoff were deemed eligible. The exclusion criteria were (1) resections without curative intent; (2) studies not reporting a specific definition of the surgical margin and cutoff used; (3) an absence of follow-up data according to margin status; (4) conference abstracts, case reports, and letters to the editors. Studies written in a language other than English were also excluded. The study protocol was registered within the PROSPERO database (registration number: ID CRD42024545496).

### 2.2. Data Extraction

Following the identification of eligible studies, the abstracts and full texts were selected by the two authors. The reference lists of the retrieved articles were screened to find additional studies not identified through the original search. The entire text of the screened papers and their eligibility were made independently by two authors (A.A.F. and F.S.). Any disagreement was solved through discussion and reassessment of the data by all the authors. One author (D.A.) extracted the data in a standardized collection form. The collected variables included the following: patient demographics (age, gender, and history of hepatitis B or C [HBV or HCV]), baseline clinical and biologic characteristics (liver cirrhosis, Child–Pugh score, and alpha-fetoprotein [AFP] level), tumor characteristics (size, number, differentiation, and microvascular invasion), operative data, and long-term outcomes. Among these variables, a special focus was reserved to anatomical resection (AR) vs. non-anatomical resection (NAR), resection margin (R0/R1, vascular, and parenchymal), and the cutoff used to define the margin status and outcomes (overall survival [OS] and disease-free survival [DFS]). The extracted data were incorporated into tables and analyzed cumulatively when possible.

Given the heterogeneity of the included studies in terms of the design, patient characteristics, and outcome measures, a meta-analysis was not feasible. Instead, we employed a structured narrative synthesis approach. The subgroup-specific findings—such as those related to MVI status, AFP levels, cirrhosis, tumor size, and type of resection—were extracted and analyzed separately when reported. This approach aligns with the PRISMA 2020 guidance for non-quantitative synthesis and allows for clinically relevant insights to be presented, even in the absence of homogeneous data amenable to meta-analysis.

### 2.3. Reported Analysis and Bias Assessment

The data were summarized and analyzed with descriptive statistics. The long-term outcomes were reported according to the margin cutoff separately for DFS and OS. The quality of studies included in this systematic review was scored by two researchers using the modified Newcastle–Ottawa scale (NOS) (with a score ranging from 0 to 9 points). The NOS is a review tool for evaluating the risk of bias in observational studies. The scale consists of four domains of risk of bias assessment: (i) selection bias; (ii) performance bias; (iii) detection bias; and (iv) information bias [16]. The overall score was converted to Agency for Healthcare and Quality standards according to the number of stars for each item.

The graphical abstract of this manuscript was made using ChatGPT 4o.

## 3. Results

The systematic review initially identified 254 records from the database searches. Before the screening, twenty-three duplicate records were removed, along with three records for other reasons. This left 228 records for screening. Of these, 168 were excluded, and 60 reports were sought for retrieval. Two reports could not be retrieved, leaving fifty-eight reports assessed for eligibility. Of these, 10 reports were excluded for reasons including duplication (*n* = 3), being systematic reviews or meta-analyses (*n* = 3), and lacking a definition of margin (*n* = 4). Ultimately, 48 studies were included in the final review. The PRISMA flow diagram showing the entire screening protocol is shown in Figure 1. Table 1 shows the overall details of the included studies and Table 2 shows the quality assessment according to the Newcastle–Ottawa scale and conversion to Agency for Healthcare and Quality standards.

### 3.1. Definition of Surgical Margins

The analysis of the included studies reveals significant variations in how the optimal resection margin for HCC is defined in the literature. While surgical resection generally determines the margin status based on a specific cutoff, in HCC, this determination is influenced by the type of resection performed and potentially by the tumor’s aggressiveness. Some studies assess margin thresholds based on whether the resection is anatomical or non-anatomical [23,48,51,52]. In contrast, other research examines margin length in relation to tumor size, stage, or histopathological characteristics.

### 3.2. Impact of Surgical Margin on Long-Term Outcomes

Among the 48 studies included in the review, 36 examined the correlation between resection margin width and OS, while 42 evaluated its association with DFS. Various surgical margin cutoffs were analyzed across the studies, including 20 mm, 10 mm, 5 mm, 4 mm, 2 mm, and 1 mm. Table 3 provides an overview of the margin status influence on DFS and OS based on the margin width. Some studies are reported more than once when they evaluated multiple surgical margin groups separately (e.g., both 5 mm and 10 mm). Specifically, four studies (Shi M. et al., 2007 [49]; Lee K.T. et al., 2012 [37]; Lee J.C. et al., 2019 [39]; Shapera et al., 2023 [47]) contributed data to more than one margin category, resulting in 52 entries from 48 included studies.

#### 3.2.1. Surgical Margin: 20 mm

Regarding overall survival (OS), two studies established a resection margin cutoff of 20 mm [26,49]. The authors reported that a 20 mm margin was associated with improved OS compared to a 10 mm margin, particularly in cases of solitary HCC ≤ 20 mm [49]. These studies also analyzed the impact of margin width on recurrence risk following liver resection, concluding that a 20 mm margin was similarly linked to better disease-free survival (DFS) compared to a 10 mm cutoff.

#### 3.2.2. Surgical Margin: 10 mm

A total of 31 studies assessed the correlation between a 10 mm free surgical margin and OS (Table 2) [17,18,19,20,21,26,28,29,30,34,35,36,37,38,40,41,43,44,45,47,48,49,53,54,55,56,58,59,60,63,64]. Among these, 11 studies found no significant association between the resection margin width and OS when using this cutoff [18,19,36,37,38,40,43,44,45,54,58,60,63,64]. Conversely, 15 studies reported that a surgical margin greater than 10 mm was linked to improved OS [17,20,21,28,29,30,34,35,41,43,48,53,55,58,60] and, in 11 cases, it was identified as an independent prognostic factor for survival [20,21,28,29,30,35,41,43,48,58,60]. Three studies compared the 10 mm margin width with smaller (1 mm and 1–9.9 mm) [47] and larger (20 mm) [49,56] cutoffs, demonstrating that wider margins were associated with better survival outcomes. However, two studies indicated that the independent correlation between a margin width of ≥10 mm and OS was only observed in the presence of microvascular invasion (MVI) [48,59]. In contrast, Han et al. reported that a narrow resection margin was associated with worse OS regardless of MVI status [26]. Additionally, Park et al. found that a 10 mm cutoff was correlated with improved OS only in patients with 1^8^F-FDG PET-positive HCC [44].

A total of 33 studies assessed the correlation between a 10 mm surgical margin and DFS (Table 2) [17,18,19,20,26,27,29,30,34,36,37,38,39,40,41,42,43,44,45,46,48,49,50,51,52,55,58,59,60,61,62,63,64]. Of these, 25 found that a surgical margin > 10 mm was associated with improved DFS [17,18,19,26,29,30,34,36,37,38,39,40,41,42,43,46,48,49,50,55,58,59,60,61,62] and, in 18 cases, this cutoff was identified as an independent predictor of recurrence [18,19,26,29,30,34,36,37,41,42,43,46,48,50,55,58,61,62]. This association was observed in various contexts, including HBV- and HCV-related HCC [46,60,61], young patients [60], both cirrhotic [18] and non-cirrhotic livers [20], in early-stage HCC [17,30,42,46,49,56,62], and BCLC B-C HCC [19].

Nevertheless, some authors found that a free surgical margin >10 mm is only necessary in more aggressive HCC, defined as in preoperative circulating tumor cells >1 [63] for patients with a baseline AFP > 200 ng/mL [39] and in cases with MVI+ [27,51,59]. Furthermore, Shi et al. reported that in this last specific subgroup of HCC, a free resection margin of 10 mm is not sufficient to improve DFS, but an AR should also be performed [48]. Regarding the influence of the resection margin on outcomes in anatomic or non-anatomic resection, two studies reported that a significative correlation was confirmed only when performing an NAR [52,55]. Finally, five articles reported no significative difference when comparing patients with a free margin ≤10 mm and >10 mm [20,44,45,51,64].

#### 3.2.3. Surgical Margin: 5 mm

Among the 48 studies, four articles compared the OS and DFS rates using a 5 mm free surgical margin [24,25,31,37], while two studies examined this cutoff solely in relation to recurrence risk [23,39]. Of these, two studies found no significant association between a 5 mm surgical margin and improved OS or DFS [25,31]. Similarly, Lee et al. reported no differences in survival or recurrence rates when comparing 5 mm and 10 mm margin cutoffs [37]. Only two studies indicated that a surgical margin greater than 5 mm was necessary for improved long-term outcomes, but this benefit was observed only in specific patient subgroups—those with a high alpha-fetoprotein tumor burden score (ATS) [24] and those with an AFP level between 15 and 200 ng/mL [39].

Additionally, only one study examined the relationship between surgical margin status and the type of hepatic resection, concluding that a margin wider than 5 mm is essential for improved DFS when a non-anatomical resection (NAR) is performed [23].

#### 3.2.4. Surgical Margin: 4 mm

One article set the cutoff surgical margin at 4 mm finding no significant differences in the OS and DFS curves between the R0 and R1 cases [32].

#### 3.2.5. Surgical Margin: 2 mm

One article assessed the influence of a 2 mm surgical margin and the long-term outcomes of patients undergoing liver resection for early-stage solitary HCC (<5 cm), finding an independent correlation between OS and DFS and a negative surgical margin (>2 mm) [57]. This association was, in the subgroup analysis, confirmed only for those cases showing MVI and no cirrhosis.

#### 3.2.6. Surgical Margin: 1 mm

Two articles assessed the surgical margin cutoff of 1 mm in terms of survival rate [33,47]. In one article, no significant difference was found in terms of OS between the R0 and R1 cases [33]. Shapera et al. compared the survival curves in cases ≤1 mm, between 1.1 and 9.9 mm and ≥10 mm, reporting improved survival rates for larger cutoffs [47]. Two authors evaluated the rate of recurrence according to a surgical margin width of 1 mm, reporting no statistically differences on DFS between the R1 and R0 patients [22,33].

### 3.3. Influence of Specific Positive Margin Cutoffs on Specific Patterns of Recurrence

Beyond the influence of recurrence and survival rates according to the margin status, the included articles sometimes reported how the margin cutoff assessed can be associated with a specific pattern of recurrence. The patterns evaluated were related to (i) timing (early versus late recurrence), (ii) site (intrahepatic versus extrahepatic), and iii) intrahepatic site (marginal versus distal intrahepatic). Fifteen articles focused on the timing of recurrence according to a margin threshold [22,24,25,26,29,37,42,43,45,49,51,55,57,59,63]. Eight of them showed a significantly lower rate of early recurrence in their corresponding wider resection margin group, which was mostly set at 10 mm [26,29,42,43,59,63], while, for two other papers, the cutoff used was 5 mm [24] and 2 mm [57]. Seven authors, on the contrary, described no significant difference in terms of the timing of recurrence between their corresponding resection margin groups when assessing the margin status [22,25,37,45,49,51,55]. The resection margin length in this case was more heterogeneous.

The second subtype of recurrence pattern assessed is intrahepatic versus extrahepatic. Nine articles reviewed the relationship between the site of disease relapse and the margin status, but no paper showed a significant difference in relation to a specific cutoff [25,31,33,37,38,40,43,45,55].

Finally, nine articles assessed the distance of the intrahepatic recurrence from the resection margin, according to the margin status in the resected specimen [22,25,31,33,37,38,43,45,49]. The definition of “marginal” recurrence varied among these studies; however, a distance of 1 or 2 cm from the resection margin was adopted by the authors. Five of these papers [22,25,33,37,45] showed no significant difference in the marginal recurrence rate, whereas the remaining four [31,38,43,49] showed a higher proportion of marginal recurrence in the case of narrow margin groups including margins of <10 mm and <5 mm.

## 4. Discussion

Surgical resection is the primary treatment option for HCC, aiming to achieve complete tumor removal while preserving a sufficient future liver remnant. For decades, there has been ongoing debate and a lack of consensus regarding the optimal resection margin for HCC [45,65,66,67]. In surgical oncology, the resection margin width is a critical factor, as it directly influences the oncological outcomes, including the tumor recurrence risk and OS. In the case of HCC, which often coexists with underlying liver disease, surgeons must carefully balance the need for a wider margin with the necessity of preserving adequate liver function.

Traditionally, a resection margin of at least 1 cm—or even 2 cm in some studies—has been considered necessary to achieve optimal outcomes in HCC surgery [56,68]. However, recent research has challenged this perspective, suggesting that a smaller margin may be equally effective in ensuring favorable long-term outcomes for HCC patients [22].

The number of randomized studies on resection margin width in HCC is very limited, with most available research consisting of retrospective cohort or case-controlled studies. Among the studies included in this review, only one was a randomized trial, conducted by Shi M. et al., which categorized patients with solitary HCC into two groups: those with a wide resection margin (2 cm) and those with a narrow resection margin (1 cm) [49]. The primary endpoint was achieved only in patients with HCC tumors ≤ 2 cm in size.

Similarly, none of the studies investigating the effects of AR were randomized to eliminate potential biases. Anatomical liver resection was first described by Makuuchi et al. in 1985 and involves resecting the tumor along with the corresponding liver segment or subsegment, including the tumor-bearing portal tributaries [69]. HCC cells can infiltrate arterioportal shunts within the tumor, spread through the portal system, and seed the adjacent liver parenchyma. Based on this mechanism, it has been hypothesized that anatomical resection may be an effective technique for achieving complete tumor removal, including micrometastases [70].

This approach has also been applied to intrahepatic cholangiocarcinoma, where it has shown promising results in selected cases [71]. While some studies included in this review suggested the potential benefits of ALR in improving RFS and OS, it is important to note that these findings were based on retrospective data or limited single-center experiences. As a result, the generalizability of these results remains uncertain.

It is important to acknowledge the high degree of clinical and methodological heterogeneity among the studies included in this review. The diversity in patient populations, tumor biology (e.g., MVI and AFP levels), underlying liver conditions (e.g., cirrhosis vs. non-cirrhosis), and surgical approaches (AR vs. NAR) limits the ability to draw uniform conclusions. Therefore, this present review does not attempt to determine a single optimal margin cutoff for all patients but rather emphasizes the need for individualized surgical strategies. This heterogeneity reinforces the rationale for a precision medicine approach, tailoring resection margins to tumor-specific and patient-specific factors.

Conducting randomized trials on resection margins in HCC presents significant ethical, clinical, and logistical challenges. Ethically, randomizing patients to specific margin widths could compromise outcomes, particularly when wider margins are considered safer for individuals with multiple high-risk factors for recurrence. Clinically, the need to balance oncologic effectiveness with liver function preservation further complicates standardizing the margin allocation. From a logistical perspective, implementing such trials is highly demanding due to the complexity of surgical planning and patient selection criteria.

Finally, the main finding of this systematic review probably lies in the absence of significant differences in the outcomes between the included groups, when focusing on the margins alone without analyzing the other multiple concomitant high-risk factors, as mentioned earlier. For HCC, tumor biology plays a pivotal role in defining the diagnosis, the prognosis, and the response to treatments. Indeed, the most significant part of this study shows how the impact of different margins’ widths varies with other factors such as the presence of MVI, the tumoral grading, the underlying liver condition, etc. Endo Y et al. found that, in patients with a high alpha-fetoprotein tumor burden score, a wider resection margin was associated with incrementally better OS and RFS [24]. Similarly, Lee JC et al. published their results revealing the need of larger resection margins for high AFP lesions [39]. Yang P et al. demonstrated a better 5-year RFS and OS among patients with wide resection margins and having MVI [59]. This was demonstrated as well by Wang et al., Han J et al., and Hirokawa F et al. in the case of solitary HCC [26,27,57]. Interestingly, Tsilimigras DI et al. found that wider margins were more important among patients undergoing non-anatomic liver resections for <5cm T1 HCC [55]. We know that a PET FDG CT scan is not a good modality for HCC in general; however, Park JH et al. showed that a resection margin size > 1 cm may improve OS in patients with PET-positive HCC [44]. Zhou KQ et al. concluded in their study that a surgical margin of >1 cm should be achieved for patients with positive circulating tumor cells [63].

Practical recommendations for surgical margin according to patient and tumor Characteristics: although the overall evidence remains heterogeneous, several subgroup analyses suggest that specific patient and tumor characteristics may benefit from wider resection margins.

Small solitary tumors (≤2 cm): in patients with early-stage solitary HCC, achieving a resection margin greater than 10 mm appears to significantly improve disease-free survival, as demonstrated by Shi M. et al. and others [49].

Microvascular invasion (MVI): The presence of MVI has been associated with poorer oncological outcomes. The studies by Yang P. et al. [59] and Shi F. et al. [48] indicate that, in MVI-positive patients, achieving a margin >10 mm and favoring anatomical resection may improve both DFS and OS.

High alpha-fetoprotein (AFP) levels: Elevated AFP levels (>200 ng/mL) have been correlated with a worse prognosis. Wider resection margins have been shown to mitigate this risk, especially among patients with high AFP tumor burden scores, as noted by Endo Y. et al. [24] and Lee J.C. et al. [39].

Non-cirrhotic liver background: in patients with preserved liver function without cirrhosis, wider margins are more easily achievable and have been associated with better oncological outcomes, according to Wang H. et al. [57].

Therefore, while a universal margin cutoff cannot be defined, a more tailored approach to margin selection based on these risk factors is advisable to optimize postoperative outcomes.

In this light, the concept of precision medicine may play a crucial role in redefining the ideal resection margin for HCC. By considering the tumor biology and characteristics of the individual patient, surgeons can tune the surgical approach to achieve the best possible outcome. Advances in imaging technology, such as preoperative imaging and intraoperative ultrasound, have improved the ability to accurately assess the extent of the tumor and guide the surgeon in determining the optimal resection margin.

## 5. Conclusions

The optimal resection margin for hepatocellular carcinoma remains variable and must be individualized based on specific tumor and patient factors. Subgroup analyses suggest that wider surgical margins (>10 mm) may confer better survival outcomes, particularly in patients with solitary small tumors (≤2 cm), microvascular invasion, high AFP levels, and a non-cirrhotic liver background. Surgical decision-making should, therefore, integrate tumor biology, liver function, and patient-specific risk profiles, rather than relying solely on arbitrary margin thresholds. Further prospective studies are needed to validate these subgroup findings and to refine the margin strategies for personalized HCC management.

## Figures and Tables

**Figure 1 cancers-17-01759-f001:**
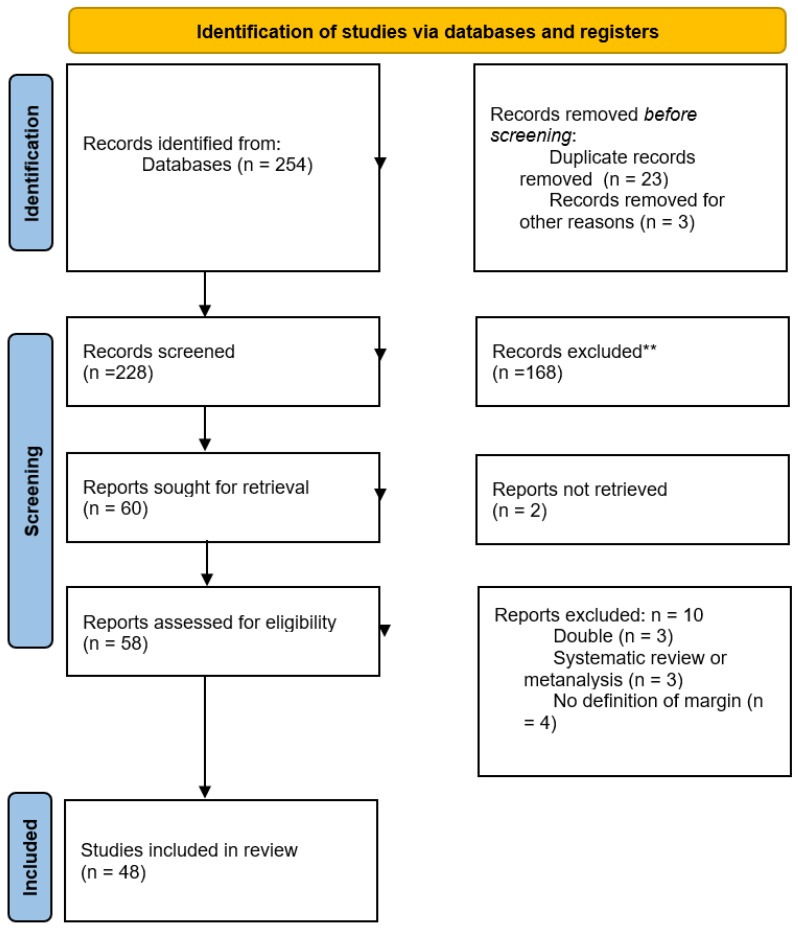
The PRISMA flow diagram. ** If automation tools were used, indicate how many records were excluded by a human and how many were excluded by automation tools. Source: [15] Page MJ, et al. BMJ 2021;372:n71. doi: 10.1136/bmj.n71. This work is licensed under CC BY 4.0. To view a copy of this license, visit https://creativecommons.org/licenses/by/4.0/ (accessed on 22 May 2025).

**Table 1 cancers-17-01759-t001:** Overall details.

Study	Study Type	Study Country	Patients, n°	Inclusion Period	Margin Assessed, mm	Type HCC	Type Resection	Underlying Conditions
Bai S. et al., 2023 [17]	Retrospective	China	670	2016–2017	10	-	With/without adjuvant TACE	-
Belli G. et al., 2011 [18]	Retrospective	Italy	109	2000–2008	10	-	Laparoscopic	Cirrhosis
Chang W.T. et al., 2012 [19]	Retrospective	Taiwan	478	1991–2006	10	BCLC B-C	-	-
Chen M.F. et al., 2003 [20]	Retrospective	Taiwan	254	1986–1998	10	-	-	Non-cirrhotic liver
Chen Z.H. et al., 2021 [21]	Retrospective	Multicentric, China	1.517	2009–2012	10	MVI+	-	-
Cheng C.H. et al., 2022 [22]	Retrospective	Taiwan	983	2003–2009	1	-	-	-
Dong S. et al., 2016 [23]	Retrospective	China	586	2001–2012	5	Solitary, without macroscopical vascular invasion	-	-
Endo Y. et al., 2023 [24]	Retrospective	Multicentric	782	2000–2020	5	-	-	-
Field W.B.S. et al., 2017 [25]	Retrospective	USA	3.300	2002–2016	5	-	-	-
Han J. et al., 2019 [26]	Retrospective	Multicentric, China	801	2007–2016	10	Solitary HCC	-	-
Hirokawa F. et al., 2014 [27]	Retrospective	Japan	167	2000–2010	10	Solitary HCC	-	-
HsiaoJ.H. et al., 2017 [28]	Retrospective	Taiwan	221	2006–2014	10	-	With/without adjuvant TACE	-
Huang G. et al., 2013 [29]	Retrospective	China	1.040	2006–2008	10	-	-	High baseline HBV-DNA
Huang W.J. et al., 2015 [30]	Retrospective	Taiwan	230	2007–2009	10	Stage I HCC	-	-
Jeng K.S. et al., 2013 [31]	Retrospective	Taiwan	196	1994–2010	5	Centrally located HCC	-	-
Ke Q. et al., 2023 [32]	Retrospective	Multicentric, China	1.033	2012–2015	4	Solitary HCC	AR	-
Kobayashi N. et al., 2020 [33]	Retrospective	Japan	454	2001–2012	1	Solitary HCC	-	-
Laurent C. et al., 2005 [34]	Retrospective	France	108	1985–2002	10	-	-	Non-cirrhotic liver
Lee. C.S. et al., 1996 [35]	Retrospective	Taiwan	48	1979–1984	10	Small asymptomatic HCC	-	-
Lee S.G. et al., 2006 [36]	Retrospective	Korea	100	1997–2003	10	Huge HCC	-	-
Lee K.T. et al., 2012 [37]	Retrospective	Taiwan	407	2000–2007	1–5 6–10 >10	-	-	-
Lee W. et al., 2018 [38]	Retrospective	South Korea	419	2004–2014	10	-	-	-
Lee J.C. et al., 2019 [39]	Retrospective	Taiwan	534	2003–2007	<5 5–9 ≥10	-	-	-
Lim C. et al., 2020 [40]	Retrospective	Multicentric, France, and Spain	187	2007–2016	10	Transplantable HCC	-	Cirrhosis
Lise M. et al., 1998 [41]	Retrospective	Italy	100	1977–1995	10	-	-	-
Liu Y. et al., 2016 [42]	Retrospective	China	223	2004–2011	10	-	-	-
Liu L. et al., 2021 [43]	Retrospective	China	240	2014–2016	10	-	-	-
Park J.H. et al., 2018 [44]	Retrospective	Korea	92	2012–2015	10	-	-	-
Poon R.T.P. et al., 2000 [45]	Retrospective	China	288	1989–1997	10	-	-	-
Sasaki Y. et al., 2006 [46]	Retrospective	Japan	417	1990–1999	10	-	-	HBV-or HCV-related HCC
Shapera E. et al., 2023 [47]	Retrospective	USA	58	2016–2022	≤1 1.1–9.9 ≥10	-	-	-
Shi F. et al., 2019 [48]	Retrospective	Japan	276	2006–2015	10	Early HCC	-	-
Shi M. et al., 2007 [49]	Prospective Randomized Trial	China	169	1999–2003	20 vs. 10	Solitary HCC	-	-
Shimada K. et al., 2008 [50]	Retrospective	Japan	117	1996–2003	10	Small HCC eligible for percutaneous local ablative therapy *	-	-
Shin S. et al., 2018 [51]	Retrospective	Korea	116	2006–2015	10	Solitary < 3 cm	-	-
Su C.M. et al., 2021 [52]	Retrospective	Taiwan	159	1997–2017	10	Solitary < 2 cm	-	CP A *
Takano S. et al., 2000 [53]	Retrospective	Japan	300	1987–1997	10	-	-	-
Torii A. et al., 1993 [54]	Retrospective	Japan	59	1981–1991	10	Solitary ≤ 3 cm	Minor/Major resection **	-
Tsilimigras D. et al., 2020 [55]	Retrospective	Multicentric	404	1998–2017	10	T1 HCC	-	-
Wang J. et al., 2010 [56]	Retrospective	China	438	1991–2004	20 vs. 10			
Wang H. et al., 2020 [57]	Retrospective	China	904	2009–2010	2	Solitary HCC ≤ 5 cm	-	-
Yang J. et al., 2014 [58]	Retrospective	China	1.084	2006–2012	10	-	-	-
Yang P. et al., 2019 [59]	Retrospective	China	2.508	2000–2013	10	-	-	HBV-related HCC
Zeng J. et al., 2020 [60]	Retrospective	China	699	2008–2015	10	-	-	HBV-related HCC, patients ≤ 40 years-old
Zhang X.F. et al., 2014 [61]	Retrospective	China	302	2008–2011	10	-	-	HBV-related HCC
Zhang H. et al., 2022 [62]	Retrospective	China	425	2015–2018	10	-	Laparoscopic	-
Zhou K.Q. et al., 2020 [63]	Retrospective	China	309	2010–2015	10	-	-	-
Zhou Z. et al., 2021 [64]	Retrospective	China	217	-	10	Solitary HCC	-	-

HCC: Hepatocellular carcinoma; TACE: transarterial chemoembolization; BCLC: Barcelona Clinic Liver Cancer prognosis and treatment strategy; MVI: microvascular invasion; HBV: hepatitis B virus; HCV: hepatitis C virus; DNA: deoxyribonucleic acid; AR: anatomical resection; and CP: Child–Pugh. * Eligible or percutaneous local ablative therapy: the criteria for local ablation therapy was up to three nodules 30 mm in size. ** Minor/Major, resection of </> three or more segments.

**Table 2 cancers-17-01759-t002:** Newcastle–Ottawa quality assessment.

Study	Selection	Sample Size	Detection	Confounding	Detection
Bai S. et al., 2023 [17]	High	High	Low	Low	Unclear/High
Belli G. et al., 2011 [18]	High	High	Low	High	Unclear/High
Chang W.T. et al., 2012 [19]	Moderate	High	Low	High	Unclear/High
Chen M.F. et al., 2003 [20]	High	High	Low	High	Unclear/High
Chen Z.H. et al., 2021 [21]	High	High	Low	High	Unclear/High
Cheng C.H. et al., 2022 [22]	Moderate	High	Low	Low	Unclear/High
Dong S. et al., 2016 [23]	High	High	Low	High	Unclear/High
Endo Y. et al., 2023 [24]	High	High	Low	High	Unclear/High
Field W.B.S. et al., 2017 [25]	High	High	Low	High	Unclear/High
Han J. et al., 2019 [26]	High	High	Low	High	Unclear/High
Hirokawa F. et al., 2014 [27]	High	High	Low	High	Unclear/High
Hsiao J.H. et al., 2017 [28]	High	High	Low	High	Unclear/High
Huang G. et al., 2013 [29]	High	High	Low	High	Unclear/High
Huang W.J. et al., 2015 [30]	High	High	Low	High	Unclear/High
Jeng K.S. et al., 2013 [31]	High	High	Low	High	Unclear/High
Ke Q. et al., 2023 [32]	High	High	Low	Low	Unclear/High
Kobayashi N. et al., 2020 [33]	Moderate	High	Low	Low	Unclear/High
Laurent C. et al., 2005 [34]	Moderate	High	Low	High	Unclear/High
Lee. C.S. et al., 1996 [35]	High	High	Low	High	Unclear/High
Lee S.G. et al., 2007 [36]	High	High	Low	High	Unclear/High
Lee K.T. et al., 2012 [37]	Moderate	High	Low	High	Unclear/High
Lee W. et al., 2018 [38]	High	High	Low	High	Unclear/High
Lee J.C. et al., 2019 [39]	Moderate	High	Low	High	Unclear/High
Lim C. et al., 2020 [40]	Moderate	High	Low	High	Unclear/High
Lise M. et al., 1998 [41]	Moderate	High	Low	High	Unclear/High
Liu Y. et al., 2016 [42]	High	High	Low	High	Unclear/High
Liu L. et al., 2021 [43]	High	High	Low	High	Unclear/High
Park J.H. et al., 2018 [44]	High	High	Low	High	Unclear/High
Poon R.T.P. et al., 2000 [45]	High	High	Low	High	Unclear/High
Sasaki Y. et al., 2006 [46]	High	High	Low	High	Unclear/High
Shapera E. et al., 2023 [47]	Moderate	High	Low	High	Unclear/High
Shi F. et al., 2019 [48]	High	High	Low	High	Unclear/High
Shi M. et al., 2007 [49]	Low	Low	Low	High	Unclear/High
Shimada K. et al., 2008 [50]	High	High	Low	High	Unclear/High
Shin S. et al., 2018 [51]	High	High	Low	High	Unclear/High
Su C.M. et al., 2021 [52]	Moderate	High	Low	High	Moderate
Takano S. et al., 2000 [53]	High	High	Low	High	Unclear/High
Torii A. et al., 1993 [54]	High	High	Low	High	Unclear/High
Tsilimigras D. et al., 2020 [55]	High	High	Low	High	Unclear/High
Wang J. et al., 2010 [56]	High	High	Low	High	Unclear/High
Wang H. et al., 2020 [57]	Moderate	High	Low	Low	Unclear/High
Yang J. et al., 2014 [58]	High	High	Low	High	Unclear/High
Yang P. et al., 2019 [59]	Moderate	High	Low	Low	Unclear/High
Zeng J. et al., 2020 [60]	High	High	Low	High	Unclear/High
Zhang X.F. et al., 2014 [61]	High	High	Low	High	Unclear/High
Zhang H. et al., 2022 [62]	High	High	Low	High	Unclear/High
Zhou K.Q. et al., 2020 [63]	Moderate	High	Low	High	Unclear/High
Zhou Z. et al., 2021 [64]	High	High	Low	Low	Unclear/High

**Table 3 cancers-17-01759-t003:** OS and DFS.

Study	Patients, n°	OS			DFS		
		Univariate Analysis, *p*-Value	Multivariate Analysis, *p*-Value	Subgroup Analysis	Univariate Analysis, *p*-Value	Multivariate Analysis, *p*-Value	Subgroup Analysis
Margin assessed = 20 mm							
Shi M et al., 2007 [49]	169	0.008	0.003	-	0.046	-	-
Wang J. et al., 2010 [56]	438	<0.001	0.011	-	-	0.014	-
Margin assessed = 10 mm							
Bai S. et al., 2023 [17]	670	0.005	-	-	0.026	-	-
Belli G. et al., 2011 [18]	109	-	-	-	0.0014	0.022	-
Chang WT et al., 2012 [19]	478	-	-	-	-	0.042	-
Chen M.F. et al., 2003 [20]	254	0.0008	-	-	0.0823	NI	-
Chen Z.H. et al., 2021 [21]	1.517	-	0.006	-	-	-	-
Han J.et al., 2019 [26]	801	<0.001	<0.001	Independent prognostic factor both in MVI+ and MVI− (*p* = < 0.001)	0.001	0.001	Independent prognostic factor both in MVI+ and MVI− (*p* = < 0.001)
Hirokawa F. et al., 2014 [27]	167	-	-	-	-	-	Significant only in MVI+ (*p* = 0.0263)
Huang G. et al., 2013 [29]	1.040	<0.001	-	-	0.001	0.001	-
Hsiao J.H. et al., 2017 [28]	221	0.0178	0.0433	-	-	-	-
Huang W.J. et al., 2015 [30]	230	<0.001	0.007	In MVI−, better RFS regardless of AR or NAR. In MVI+, AR and ≥10 mm, better RFS	<0.001	<0.001	In MVI−, better RFS regardless of AR or NAR. In MVI+, AR and ≥10 mm, better RFS
Laurent C. et al., 2005 [34]	108	0.01	-	-	0.005	0.035	-
Lee W. et al., 2018 [38]	419	0.690	-	-	0.042	0.146	-
Lee K.T. et al., 2012 [37]	407	NS	-	-	0.023	0.010	-
Lee J.C. et al., 2019 [39]	534	-	-	-	0.042	-	Significative in AFP > 200 ng/mL (*p* = 0.012)
Lee. C.-S. et al., 1996 [35]	48	0.04	0.036	-	-	-	-
Lee S.G. et al., 2007 [36]	100	0.075	-	-	0.009	0.001	-
Lim C. et al., 2020 [40]	187	0.70	-	-	0.03	-	-
Lise M. et al., 1998 [41]	100	0.04	0.05	-	0.05	0.03	-
Liu L. et al., 2021 [43]	240	<0.001	0.013	-	<0.001	0.011	-
Liu Y. et al., 2016 [42]	223	-	-	-	0.005	0.006	Analysis performed for the risk of recurrence
Park J.H. et al., 2018 [44]	92	0.117	-	Significative difference only in PET-FDG (+) HCC (*p* = <0.001)	0.302	-	Not significative both in PET-FDG (+) and (−) HCC
Poon R.T.P. et al., 2000 [45]	288	0.495	-	-	0.943	NI	-
Sasaki Y. et al., 2006 [46]	406	-	-	-	0.002	0.049	-
Shapera E. et al., 2023 [47]	58	0.013	-	-	-	-	-
Shi M. et al., 2007 [49]	169	0.008	0.003	-	0.046	-	
Shi F. et al., 2019 [48]	276	<0.001	0.007	RM > 10 mm independent from AR/NAR in MVI− patients. In MVI+, both RM > 10 mm and AR are necessary	<0.001	<0.001	RM > 10 mm independent from AR/NAR in MVI− patients. In MVI+, both RM > 10 mm and AR are necessary
Shimada K. et al., 2008 [50]	117	-	-	-	0.0203	0.034	-
Shin S. et al., 2018 [51]	116	-	-	-	0.453	-	Suggested RM > 1 cm in MVI+ (*p* = 0.049)
Su C.M. et al., 2021 [52]	159	0.053	-	-	0.096	-	-
Takano S. et al., 2000 [53]	300	0.0125	-	-	-	-	-
Torii A. et al., 1993 [54]	59	<0.1	0.7191	-	-	-	-
Tsilimigras D.I. et al., 2020 [55]	404	0.047	-	-	0.02	0.017	In AR, the RM is not an independent risk factor. In NAR, the RM is an independent risk factor
Yang J. et al., 2014 [58]	1.084	0.005	0.002	-	0.007	0.011	-
Yang P. et al., 2019 [59]	2.508	<0.001	-	Independent prognostic factor in MVI+ (*p* ≤ 0.001)	<0.001	-	Independent prognostic factor in MVI+ (*p* ≤ 0.001)
Zeng J. et al., 2020 [60]	699	<0.01	<0.01	-	<0.01	-	-
Zhang X.F. et al., 2014 [61]	302	-	-	-	0.048	0.048	-
Zhang H. et al., 2022 [62]	425	-	-	-	0.019	0.002	-
Zhou K.Q. et al., 2020 [63]	309	-	-	Not significative in CTC > 1 (*p* = 0.078)	-	-	Independent risk factor when CTC > 1 (*p* ≤ 0.023)
Zhou Z. et al., 2021 [64]	817	0.067	-	-	>0.05	-	-
Margin assessed = 5 mm							
Dong S. et al., 2016 [23]	586	-	-	-	0.000	0.001	Suggests in NAR an RM > 5 mm (*p* ≤ 0.05)
Endo Y. et al., 2023 [24]	782	<0.001	<0.01	Especially with a high alpha-fetoprotein tumor burden score (ATS) (*p* ≤ 0.05)	NI	NI	Especially with a high alpha-fetoprotein tumor burden score (ATS) (*p* ≤ 0.05)
Field W.B.S. et al., 2017 [25]	3300	0.23	-	-	0.33	-	-
Lee K.T. et al., 2012 [37]	407	NS	-	-	0.320	0.457	-
Lee J.C. et al., 2019 [39]	534	-	-	-	0.027	0.024	Significative in AFP > 200 ng/mL (*p* = 0.012)
Jeng K.-S. et al., 2013 [31]	196	0.055	-	-	0.066	-	-
Margin assessed = 4 mm							
Ke Q. et al., 2023 [32]	1.033	0.150	-	-	0.470	-	-
Margin assessed = 2 mm							
Wang H. et al., 2020 [57]	904	<0.001	<0.001	Significative in MVI+ (*p* = 0.001) and in non-cirrhotic (*p* = 0.001)	<0.001	<0.001	Significative in MVI+ (*p* ≤ 0.001) and in non-cirrhotic (*p* ≤ 0.001)
Margin assessed = 1 mm							
Cheng C.H. et al., 2022 [22]	983	-	-	-	0.155	-	-
Kobayashi N. et al., 2020 [33]	454	0.496	-	-	0.375	-	-
Shapera E. et al., 2023 [47]	58	NI	-	-	-	-	-

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
