# Peer review of "The Definition of the Best Margin Cutoff and Related Oncological Outcomes After Liver Resection for Hepatocellular Carcinoma: A Systematic Review"

_cancers, 2025, doi:10.3390/cancers17111759_

Round 1
Reviewer 1 Report (Previous Reviewer 2)
Comments and Suggestions for Authors
Thank you for your answers. As far I am concerned the paper may be accepted for publication
Reviewer 2 Report (Previous Reviewer 3)
Comments and Suggestions for Authors
Follow up comments on the best surgical margin for operation of HCC.
I appreciated this revered version of the manuscript.
In the previous version of the manuscript, the authors concluded that the practical way to determine the best margin for operation of HCC is ‘case by case’. I did not agree with this result and suggested that the author review the articles more carefully.
Actually, heterogeneity of the hepatocellular carcinoma makes judgement to select best margin for surgical intervention for most cases difficult. The authors find, under some circumstances, there are evidence-based guidelines to choose best surgical margin. This conclusion will be useful for clinical practice.
Reviewer 3 Report (New Reviewer)
Comments and Suggestions for Authors
Interesting review of the literature.
The manuscript is well structured.
The review of the original manuscript is well organized.
The tables are exhaustive.
The research was conducted with rigor.
This manuscript is a resubmission of an earlier submission. The following is a list of the peer review reports and author responses from that submission.
Round 1
Reviewer 1 Report
Comments and Suggestions for Authors
This systematic review focuses on the optimal resection margin for hepatocellular carcinoma (HCC) surgery. While traditionally a margin of at least 1–2 cm has been considered ideal for minimizing recurrence and improving overall survival (OS), recent studies suggest that narrower margins may be equally effective in selected patients. However, most existing evidence is derived from retrospective or single-center studies, with only one randomized controlled trial identified, limiting the strength of the conclusions.
Anatomical liver resection (ALR), which targets tumor-bearing portal territories, is also discussed as a potentially superior method for removing microscopic tumor spread. Nevertheless, evidence supporting its advantage remains inconclusive due to a lack of randomized trials.
The review highlights that tumor biology—such as microvascular invasion (MVI), tumor grade, alpha-fetoprotein (AFP) levels, and liver function—plays a crucial role in determining surgical outcomes. Wider resection margins may benefit patients with high-risk features such as MVI or elevated AFP levels. Thus, a one-size-fits-all approach is inadequate.
Given the ethical, clinical, and logistical challenges of conducting randomized trials in this area, the authors advocate for a precision medicine approach. By tailoring surgical strategies to individual tumor characteristics and utilizing advanced imaging techniques, optimal outcomes may be achieved.
This paper is significant in that it clarifies various issues in the consideration of resection margins for hepatocellular carcinoma. However, although many retrospective studies were compiled, the background factors are so different that the structure of the article needs to be reexamined in terms of a systematic review.
Author Response
Reviewer Comment: "Although many retrospective studies were compiled, the background factors are so different that the structure of the article needs to be reexamined in terms of a systematic review."
Author Response: We thank the reviewer for this valuable comment. We agree that the variability among the included studies in terms of patient demographics, tumor characteristics, and surgical strategies introduces a high degree of heterogeneity, which is a limitation inherent to the current body of literature on resection margins in HCC.
To address this, we have made the following improvements:
- Discussion Section Revision: We have added a paragraph explicitly acknowledging the heterogeneity of the included studies and discussing how this impacts the generalizability and interpretability of our findings. We emphasized that while meta-analytic pooling was not feasible, structured descriptive synthesis still allows for clinical insights to be drawn when interpreted in context.
- Methods Section Update: We now include clearer justification for our narrative synthesis approach and have described how subgroup-specific findings (e.g., based on microvascular invasion status, AFP levels, presence of cirrhosis, and type of resection) were extracted and reported. This is in line with PRISMA guidelines when meta-analysis is not appropriate due to high heterogeneity.
These updates strengthen the methodological rigor and transparency of our systematic review and better reflect the complexity of surgical decision-making in HCC. We hope these revisions address the reviewer’s concerns and improve the clarity and utility of our manuscript.
Reviewer 2 Report
Comments and Suggestions for Authors
Intense work and nice presentation on a permanent hot topic related to HCC. The only comment I have is related to the specified article numbers: there are 48 articles in text, 45 on the table, and 38 or 42 when different resection margins are evaluated.
Author Response
Reviewer Comment: "Intense work and nice presentation on a permanent hot topic related to HCC. The only comment I have is related to the specified article numbers: there are 48 articles in text, 45 on the table, and 38 or 42 when different resection margins are evaluated."
Author Response:
We sincerely thank the reviewer for the encouraging feedback and for carefully noting the discrepancies in the number of studies presented. Upon detailed review, we identified the following points and have made the necessary corrections:
- In the Results section of the Abstract, there was a typing error regarding the number of studies, which has now been corrected.
- Tables 1 and 2 correctly include 48 studies, corresponding exactly to the number of studies included in the systematic review.
- Table 3 displays 52 entries because it summarizes the results according to different surgical margin cut-offs (1 mm, 2 mm, 5 mm, 10 mm, and 20 mm). In this table, four studies are listed twice as they reported separate analyses for more than one resection margin group:
- Shi M et al., 2007
- Lee K.T. et al., 2012
- Lee J.C. et al., 2019
- Shapera et al., 2023
We have added a clarifying sentence in the Methods section explaining that Table 3 includes repeated studies when multiple margin groups were evaluated within the same study.
We are grateful for the reviewer’s thorough review, which allowed us to improve the accuracy and clarity of our manuscript.
Reviewer 3 Report
Comments and Suggestions for Authors
Comments on surgical margin of HCC
It is quite a pity that after tough and thorough review of such a lot of articles, the authors cannot find clear cut criteria for the best safe surgical margin for curative operation for patients with hepatocellular carcinoma after all.
However, although most studies showed no significant difference between the patients that above and below a specific cut margin, some subgroups of patients, such as size smaller than 2 cm of larger than 5 cm and those with microvascular invasion, some articles seemed to make a little positive sense with respect to overall survival and disease free survival.
Although to create a rule that could be applied generally is not possible, I still suggested that the authors can consider to draw some valuable information from interpretation of the results of some studies and give the readers some valuable guidance for patients with specific circumstances, not merely tell us the proper management is “case by case”.
Actually, after reading this manuscript, I personally felt confused. I can not agree the authors’ conclusion saying that “By continuing to refine our understanding of the ideal resection margin for HCC, we can improve surgical outcomes and quality of life for patients undergoing HCC resection”, since this systemic review in the current version did not give the readers this kind of hope.
Author Response
Reviewer Comment: "The authors should provide more valuable and specific guidance based on the subgroup findings rather than conclude simply that management must be 'case-by-case.' Furthermore, the optimistic final statement should be reconsidered as the review, in its current version, does not strongly support it."
Author Response:
We thank the reviewer for the careful reading of our manuscript and the valuable comments provided. We fully agree with the suggestion that, despite the heterogeneity of the available evidence, some subgroup-specific findings should be highlighted to better guide clinical practice.
In response, we have revised the Discussion section by adding a new subsection titled "Practical Recommendations for Surgical Margin according to Patient and Tumor Characteristics." In this section, we now clearly present situations where wider margins (>10 mm) appear beneficial, particularly for patients with solitary small tumors (≤2 cm), microvascular invasion, high AFP levels, and non-cirrhotic livers.
Moreover, we have adjusted the Conclusions section to better reflect the nuanced findings of our review. We have removed the previous over-optimistic sentence and replaced it with a more cautious conclusion, emphasizing that wider margins should be considered selectively in high-risk patients, while a personalized surgical strategy remains essential.
Finally, we have also modified the Abstract accordingly to incorporate a brief mention of these subgroup findings, ensuring consistency throughout the manuscript.
We sincerely appreciate the reviewer’s constructive feedback, which has significantly improved the clarity and practical relevance of our work